# The Roadmap of RANKL/RANK Pathway in Cancer

**DOI:** 10.3390/cells10081978

**Published:** 2021-08-04

**Authors:** Sandra Casimiro, Guilherme Vilhais, Inês Gomes, Luis Costa

**Affiliations:** 1Luis Costa Laboratory, Instituto de Medicina Molecular—João Lobo Antunes, Faculdade de Medicina da Universidade de Lisboa, 1649-028 Lisboa, Portugal; ines.gomes@medicina.ulisboa.pt; 2Faculdade de Medicina da Universidade de Lisboa, 1649-028 Lisboa, Portugal; guilhermevilhais@campus.ul.pt; 3Oncology Division, Hospital de Santa Maria, Centro Hospitalar Universitário Lisboa Norte, 1649-028 Lisboa, Portugal

**Keywords:** bone metastasis, bone-targeted agent, breast cancer, drug repurposing, RANK ligand (RANKL), receptor activator of nuclear factor-κB (RANK), targeted therapy

## Abstract

The receptor activator of the nuclear factor-κB ligand (RANKL)/RANK signaling pathway was identified in the late 1990s and is the key mediator of bone remodeling. Targeting RANKL with the antibody denosumab is part of the standard of care for bone loss diseases, including bone metastases (BM). Over the last decade, evidence has implicated RANKL/RANK pathway in hormone and HER2-driven breast carcinogenesis and in the acquisition of molecular and phenotypic traits associated with breast cancer (BCa) aggressiveness and poor prognosis. This marked a new era in the research of the therapeutic use of RANKL inhibition in BCa. RANKL/RANK pathway is also an important immune mediator, with anti-RANKL therapy recently linked to improved response to immunotherapy in melanoma, non-small cell lung cancer (NSCLC), and renal cell carcinoma (RCC). This review summarizes and discusses the pre-clinical and clinical evidence of the relevance of the RANKL/RANK pathway in cancer biology and therapeutics, focusing on bone metastatic disease, BCa onset and progression, and immune modulation.

## 1. Introduction

Identified over two decades ago [1,2,3], the receptor activator of the nuclear factor-κB ligand (RANKL)/RANK pathway remains a hot topic in cancer research. Long studied for its role as a master regulator of osteoclastogenesis [1,4], this pathway gained renewed interest over the past decade as an important player in breast carcinogenesis [5,6,7]. More recently, compelling evidence supporting the role of the RANKL/RANK pathway in response to immunotherapy [8,9,10] fueled translational and clinical research on the biology and targeting of the RANKL/RANK pathway in cancer.

RANKL (also known as TRANCE) is a homotrimeric transmembrane protein member of the tumor necrosis factor (TNF) cytokine family, initially identified as a mediator of T cell-dependent immune response, particularly in the modulation of T cell [11] and dendritic cell (DC) activity [12]. However, it was soon found to be the same molecule as the so-called osteoclast differentiation factor (ODF), which was already known to promote functional osteoclast differentiation in presence of colony-stimulating factor 1 (CSF-1) [2,4]. Soluble RANKL derives from proteolytic cleavage by matrix metalloproteinase (MMP)-14 and A disintegrin and metalloproteinase domain-containing protein 10 (ADAM10) [13] or alternative splicing [14]. RANK is a transmembrane receptor, member of the TNF receptor (TNFR) family, activated by RANKL binding. Expressed on the surface of osteoclasts, RANK’s physiological role in bone was confirmed in vivo by the observation that RANK-deficient mice suffered from severe osteopetrosis because of impaired osteoclast differentiation [1]. The RANKL/RANK signaling axis involves a third player, osteoprotegerin (OPG), a soluble decoy receptor with a high affinity for RANKL, which is critical for bone homeostasis [15,16].

Due to its pivotal role in bone pathophysiology, efforts have been made regarding the pharmacological targeting of the RANKL/RANK pathway as a way to prevent bone resorption, with the first attempts going back to the early 2000s through the use of recombinant OPG (OPG-Fc) [17]. However, OPG-Fc clinical development was discontinued in favor of denosumab, a fully human immunoglobulin G (IgG) 2 monoclonal antibody that binds specifically to RANKL, which was later approved for the treatment of osteoporosis and cancer-induced bone metastases (BM) [18,19]. Denosumab is a very interesting example of drug repurposing, since this bone-targeted agent (BTA) is currently under investigation in other clinical settings, with promising results.

This review provides a parallel perspective of the pre-clinical and clinical evidence of RANKL/RANK pathway in cancer-induced BM, breast cancer (BCa) onset and progression, and immune modulation (Figure 1), exploring the (potential) efficacy of RANKL inhibition in all cancer stages.

## 2. The Role of RANKL/RANK Pathway in Bone Health and Disease

BM represent the most common form of distant relapse in malignancies with high incidence, as BCa and prostate cancer (PCa), and are highly prevalent in renal cell carcinoma (RCC) and multiple myeloma (MM). However, several other cancers metastasize to the skeleton. With very specific biology and pathophysiology, strictly related to unbalanced bone resorption, BM are associated with high morbidity [20,21]. BM tumor cells growing at bone metastatic niche totally subvert the physiologically balanced bone homeostasis to favor cancer spread, leading to osteolysis, skeletal-related events (SREs), and decreased overall survival (OS).

Osteoclasts and osteoblasts are the bone cells responsible for bone resorption and formation, respectively. Physiologic and pathologic osteoclastogenesis is triggered by RANKL produced by bone marrow stromal cells, osteocytes, and osteoblasts, which binds to RANK on the surface of hematopoietic osteoclast precursor cells [22]. This leads to recruitment of TNFR-associated cytoplasmic factors (TRAFs) for specific cytoplasmic RANK domains, subsequent activation of NF-κB, c-Jun N-terminal kinase (JNK), p38, extracellular signal-regulated kinase (ERK), and Src pathways, and expression of genes characteristic of active, bone-resorbing osteoclasts. CSF-1, interleukin (IL)-6, IL-8, and chemokine (C-X-C motif) ligand 12 (CXCL2) are also important regulators of osteoclastogenesis. In turn, osteoclast-mediated bone resorption releases osteoblastic factors from the bone matrix, including transforming growth factor-beta (TGF-β), bone morphogenetic proteins (BMPs), fibroblast growth factor (FGF), platelet-derived growth factor (PDGF), and insulin-like growth factors (IGFs), inducing osteoblast differentiation from stromal mesenchymal stem cells. Subsequently, OPG secreted by mature osteoblasts and stromal cells exerts a negative effect over osteoclastogenesis by sequestering RANKL.

However, metastatic tumor cells in bone express a panoply of osteoclastogenic factors that drive increased bone resorption. These include IL-1, IL-6, parathyroid hormone-related protein (PTHrP), prostaglandin E2 (PEG2), CSF-1, and TNF-alpha (TNF-α). Tumor-derived PTHrP and bone-derived IL-11 upregulate RANKL and downregulate OPG, thereby activating osteoclastogenesis. In response, exacerbated bone resorption feeds tumor cells with mitogenic factors released from the bone matrix, like TGF-β, IGFs, FGFs, PDGF, and Ca^2+^, further fueling the so-called “vicious cycle of BM” [20,23,24,25]. 

Although all BM present increased bone resorption, osteoblastic BM, as seen in PCa, also display augmented bone formation. This mostly occurs via cancer cell-produced endothelin-1 (ET-1), which stimulates endothelin A receptor(ETR) in osteoblasts, activating Wnt signaling and osteoblast activity [26].

The efficacy of RANKL inhibition in the treatment of BM was demonstrated in different in vivo models of breast, lung, prostate, and colon cancer [19,27]. Ultimately, indifferent to the radiologic nature of BM or tumor type, RANKL/RANK pathway blockade is able to abrogate cancer-induced osteoclastogenesis and bone resorption, delaying and decreasing SREs [28]. The clinical benefit of RANKL pharmacological inhibition in BM will be discussed in the next section. Yet, it is important to review the molecular and biological effects of RANKL blockade, along with evidence that rationalizes the clinical benefit of denosumab over other BTAs, like bisphosphonates (BPs). In the next sections, the impact of RANKL blockade in osteoclasts’ life span, (in)direct anti-tumor effects, and bone pre-metastatic niche modulation will be explored (Figure 2).

### 2.1. Osteoclasts

Signal transduction through RANK not only drives osteoclast differentiation from hematopoietic precursors, but also activates and prolongs the survival of mature osteoclasts [29,30]. Formation of RANKL-induced actin ring in mature osteoclasts occurs within minutes of RANK activation, and RANKL-treated mice show increased blood ionized Ca^2+^ within one hour after injection, consistently with immediate osteoclast activation in vivo [29]. Additionally, RANKL (through NFkB/JNK/Src) and CSF-1 (through NFkB/bcl-2) are required for optimal osteoclast survival [30]. 

Conversely, BPs—structural analogs of pyrophosphates with high affinity to hydroxyapatite in bone—are only effective against bone-resorbing mature osteoclasts, upon internalization [31]. BPs induce osteoclast apoptosis either by causing intracellular accumulation of adenosine triphosphate (ATP) toxic analogs (non-N-BPs, like etidronate and clodronate) or by interfering with farnesyl pyrophosphate synthase (FPPS)—a key enzyme in the mevalonate pathway and protein prenylation (N-BPs, like pamidronate, alendronate, and zoledronate [ZA]). 

Therefore, blocking RANKL/RANK pathway controls pre-osteoclast recruitment, fusion into multinucleated osteoclasts, osteoclast activation, and osteoclast survival, and this may render an advantage over BPs, which are effective in impairing the activity of mature osteoclasts but not in preventing their differentiation. This influences bone resorption rate, as evidenced by the significant decrease in urinary NTx/creatinine levels—a biomarker of bone resorption—with denosumab compared with ZA in patients with BM, independently of tumor type [32,33,34].

### 2.2. (In)Direct Anti-Tumor Effects

It is widely accepted that BTAs have an indirect anticancer effect, by decreasing bone resorption and consequently reducing the availability of matrix-trapped mitogens to fuel tumor growth. However, there is compelling pre-clinical evidence that BTAs may also directly affect metastatic tumor cells. As denosumab does not recognize murine RANKL, the vast majority of pre-clinical studies use OPG-Fc (and RANK-Fc) to mimic denosumab. 

In vitro preclinical evidence from diverse tumor types—including BCa, PCa, lung, ovarian, and bladder cancer, and RCC—suggests that BPs have direct effects over tumor cells, namely through apoptosis induction and proliferation, migration, and invasion inhibition [35,36]. Moreover, several cancer cells, including osteotropic BCa and PCa ones, express functional RANK, which induces downstream pathway activation upon RANKL stimulus, increasing migration and invasion [37,38,39,40,41], endorsing a role for RANK signaling in the acquisition of a more aggressive phenotype. However, one study in a murine BCa BM model using the osteotropic MDA-MB-231 cell line directly compared the use of ibandronate with OPG-Fc with therapeutic intents and showed no efficacy differences between either one or the combination of both [42], advocating only indirect effects of both BTAs on tumor growth. Nonetheless, RANK overexpression in the same cell line significantly increased metastatic growth rate in bone versus parental cells and, although RANKL inhibition and ZA reduced BM, RANKL inhibition was more effective, suggesting a direct effect over RANK-positive (RANK+) tumors [43]. The observation that RANKL upregulates osteotropic gene expression in cancer cells, favoring osteoclastogenesis, supported this hypothesis. Additionally, our group demonstrated that RANKL is a positive regulator of MMP-1 in MDA-MB-231 cells [39], a known osteotropic factor [44,45]. Moreover, OPG-Fc prevented tumor-induced BM in a mouse model of estrogen receptor-positive (ER+) BCa, where tamoxifen as a single agent was shown to reduce tumor growth in the hind limbs and OPG-Fc blocked bone resorption, but the combination of both was more effective [46]. The same study hypothesized that RANKL inhibition targeted the bone microenvironment and tamoxifen cancer cells. However, hawse have recently shown that RANK overexpression in ER+ BCa cells is associated with endocrine therapy (ET) resistance [47], indicating that RANKL inhibition may have increased sensitivity to tamoxifen according to results of the previous study.

In PCa, pre-clinical studies have also come to different conclusions, supporting both the “indirect-only” and “direct” anti-tumor effect hypotheses. In the LNCaP PCa mouse model, OPG-Fc had no effect in cell viability, proliferation, or basal apoptotic rate in vitro or in vivo in subcutaneous tumors, despite being effective in preventing BM development [48]. The same inefficacy in subcutaneous tumors or in vitro proliferation was reported in other studies with LNCaP [49] and LuCaP [50,51] cell lines. In vitro, RANKL expressed by LNCaP cells was shown to be osteoclastogenic [48]. It was recently reported that soluble RANKL is dispensable for physiological regulation of bone and immune systems or non-skeletal metastases, but seems to have an important role in promoting BM development by directly triggering migration of tumor cells to bone [52]. This additional evidence supports the preventive role of RANKL targeting through direct osteotropism inhibition. Accordingly, RANKL was able to trigger LNCaP [37] and PC3 [38] PCa cell migration in vitro. 

In favor of the “direct” anti-tumor effect, a recent study addressing the predictive value of RANK+ circulating cancer cells (CTCs) in metastatic BCa patients during denosumab treatment demonstrated that 70% of patients with detectable CTCs had one or more RANK+ CTCs [53]. Interestingly, whereas total baseline CTCs were associated with bone outcomes, RANK+ CTC persistence during treatment correlated with better outcomes, suggesting its relevance in RANKL inhibition efficacy. 

Overall, the direct anti-tumor effect of RANKL inhibition seems to be more deleterious to the metastatic properties of cancer cells than BPs, further contributing to the clinical benefit of BM treatment, which will be further discussed below.

### 2.3. Bone Pre-Metastatic Niche 

Apart from RANKL effects on osteoclasts and tumor cells, the role of RANKL targeting in the tumor microenvironment is also potentially relevant, extending its effects on BM development to BM prevention. RANKL was found to be an inductor of angiogenesis and increased vascular permeability in RANK-expressing endothelial cells, which may favor extravasation and metastases [54]. In this case, RANKL inhibition may also have an antiangiogenic effect, decreasing relapse in distant organs. However, the rationale for using anti-RANKL or other BTAs in adjuvant settings to prevent bone relapse relies on a decrease in bone resorption and bone-derived chemoattractant molecules, as well as on making bone a less congenial soil for cancer cell growth. 

It is acknowledged that osteotropic cancer cells use CXCR4 to “sense” CXCL12 at distant locations, including bone [55]. In addition, evidence indicates that tumor cells at the primary location can modulate the bone microenvironment as a pre-metastatic niche [56,57]. Examples include increased MAF-regulated PTHrP expression [58] and expression of the macrophage-capping protein (CAPG) and PDZ domain-containing protein GIPC1 (GIPC1) [59] in bone-tropic primary BCa. Moreover, exosomes released from PCa cells relate with BM incidence and may modulate the pre-metastatic niche [60]. Therefore, in a preventive setting, targeting both primary cancer cells and bone microenvironment may affect BM onset. 

Pre-clinical murine studies addressing RANKL inhibition in both preventive and therapeutic settings have shown that targeting RANK-expressing cancer cells not only decreases BM tumor burden but also prevents BM onset [37,61]. This suggests the combination of a less favorable pre-metastatic niche—depriving tumor cells of growth factors and cytokines that would be released from bone—and decreased cancer cell metastatic, or at least osteotropic, characteristics. The preventive effect of RANKL inhibition in BM was found to be dependent on the direct effect in RANK-mediated expression of cancer cell-derived osteotropic factors [37,38,39] and also to be more effective than BPs in tumors with high RANK expression [43]. 

According to the pathophysiology of BM, a highly resorptive post-menopausal bone would be a more attractive pre-metastatic niche for cancer cells to thrive. However, analysis of clinical series looking at disseminated tumors cells in bone marrow suggests that the longer a woman is post-menopausal, the less attractive bone microenvironment is for tumor cells [62,63]. A recent study has shown that estradiol and TGF-β upregulate PTHrP in ER+ osteotropic BCa cells [64] and increase the progression of osteolytic metastases in ER-negative BCa cells [65]. Therefore, estrogen deficiency may contribute to the benefit of adjuvant BPs in bone recurrence in post-menopausal women, while in metastatic settings, the menopausal status does not affect outcomes related to BTAs [66]. In agreement with these data, ZA reduced BM in oophorectomized mice in an animal model of BCa-induced BM [67]. However, in adjuvant setting RANKL inhibition reduced disease recurrence in bone irrespectively of menopausal status [68], which may be linked to the crosstalk between estrogen and RANKL. Estrogen is a known regulator of bone physiology and pathophysiology and has been shown to protect bone by regulating osteoclast survival, either by inducing autocrine TGF-β expression or by upregulating the apoptosis-promoting Fas ligand in osteoblasts [69,70]. Upon estrogen withdrawal, increased RANKL expression is the main mechanism underlying bone turnover upregulation. Therefore, lower RANKL levels in pre-menopausal conditions may contribute to the efficacy of RANKL inhibition. The clinical outcomes of BTAs in BM and adjuvant settings will be presented later in this review.

Recently, it has been hypothesized that the effects of BTAs in the bone microenvironment immune compartment may significantly account for pre-metastatic niche modulation [62]. BPs were shown to reduce CD11b^+^ tumor-associated macrophages (TAMs), decreasing vascular endothelial growth factor (VEGF) and MMP-9 in the tumor microenvironment and leading to anti-angiogenesis [71]. TAM reduction accompanies TAM repolarization into anti-tumor M1 macrophages [72]. Moreover, BPs can also activate cytotoxic γδT cells (Vγ9Vδ2) and promote their infiltration into tumors, which will sense IPP/ApppI-BP-induced accumulation in cancer cells as phosphoantigens [73,74]. These immunomodulatory effects of BPs have been suggested as a reason for their advantage over RANKL inhibition in preventing BM in the adjuvant setting [75]. 

Although RANKL/RANK pathway is an important immune regulator and studies have hypothesized that RANKL inhibition in patients with BM could be immunosuppressive, clinically significant effects on the immune system have not been reported in clinical trials. Evidence suggests that RANKL is an effective, but not essential, co-stimulatory factor for immune cell activation, supporting the lack of relevant clinical findings regarding the impact of RANKL inhibition in immunity. It was shown that the RANKL effect in T cells is redundant with other cytokines, and that only when major immunologically active molecules are deleted does RANKL/RANK pathway become the main co-stimulatory pathway for crosstalk between immune cells [76]. For example, although monocyte−macrophages are positively regulated by RANKL (protected from apoptosis, increased phagocytic properties, and activated antigen presentation), RANKL−/− mice have no alteration in the number and distribution of monocyte–macrophages [1], except if co-stimulatory molecules are missing (e.g., CD40L) [76]. Moreover, RANKL enhances DC survival, antigen presentation, and cytokine production in vitro, but RANK and RANKL−/− mice have intact DC development and function [1]. 

Overall, the complex crosstalk between cancer cells, bone pre-metastatic niche, and tumor microenvironment is clearly affected by RANK expression outside osteoclasts, and RANKL blockade may contribute to improved outcomes in BM patients through several complementary mechanisms. These include the indirect impact on tumor growth through a decrease in bone soil congeniality; the indirect impact mediated by anti-angiogenic properties; the direct impact in cell signaling, migration, invasion, and osteotropism; and the decreased ability of cancer cells to prepare the pre-metastatic niche in the bone. In the next section, clinical evidence on RANKL inhibition in BM will be summarized.

### 2.4. Anti-RANKL Therapy in BM: Discovery and Current and Future Perspectives 

The anti-resorptive effect of recombinant human OPG was reported two decades ago, after observations that its intravenous administration in normal rats increased bone mineral density and bone volume as a consequence of decreased active osteoclasts [17]. This was in accordance with observations that OPG-deficient mice developed early osteopenia [77]. However, despite its unequivocal physiological ability to impair bone resorption, very high subcutaneous doses (>10–30 mg/kg) of recombinant full-length OPG were required for in vivo efficacy, and its pharmacokinetic and pharmacodynamic profile was poor [78]. The best protein was a recombinant protein comprising the a.a. 22–194 of human OPG fused with the human IgG1 Fc region, found to be over 200 times more active than full-length OPG in vivo and with prolonged half-life. Nonetheless, OPG-Fc and RANK-Fc were associated with autoimmune hypercalcemia, being discontinued in favor of an anti-RANK antibody and ultimately leading to AMG 16, currently known as denosumab [19].

#### 2.4.1. Bone Metastatic Disease

Although the treatment of BM from solid tumors and MM is rarely curative, it is possible to prevent disease progression and palliate symptoms for many years using systemic anticancer treatments [20]. SREs reflect the burden of bone pain and structural damage caused by bone metastatic involvement, representing an important form of skeletal morbidity that impacts patients’ quality of life and results in significant healthcare costs [79]. SREs comprise five major complications of tumor bone disease: pathological fracture, need for radiotherapy to relieve bone pain or reduce bone structural damage, need for bone surgery to prevent or repair fractures, spinal cord compression, and hypercalcemia [20]. BTAs have been shown to improve bone structure and quality, minimizing the risk of SREs in patients with BM from solid tumors and MM [75,79]. Therefore, in order to reduce morbidity and complement other cancer-specific treatments, current clinical guidelines recommend prescribing a BTA following the initial radiological diagnosis of BM in most patients [80]. 

To understand the current clinical applications of RANKL inhibition in the context of BM, it is important to briefly review the role of BPs in the history of bone metastatic disease management. Several BPs have proven efficacious in preventing SREs in patients with BM from BCa or MM since the approval of clodronate in these indications in the early 1990s [33,81]. Still, ZA remains the only BP approved for the treatment of metastatic castration-resistant PCa (CRPC) and BM from other solid tumors [79]. The addition of a BTA in the treatment of endocrine-sensitive PCa showed no evidence of a survival improvement or SRE reduction compared with placebo and is hence not recommended outside treatment-induced bone loss prevention or pre-existing osteoporosis clinical settings [80,82]. 

Since denosumab was first licensed for the treatment of BM from solid tumors in 2010, numerous head-to-head randomized controlled trials (RCTs) have compared denosumab with ZA in bone health settings in several human cancers (Table 1) [79]. 

Denosumab was shown to be superior to ZA in delaying and preventing SREs in patients with bone metastatic BCa [32] and metastatic CRPC [33]. In an RCT including patients with BM from solid tumors and MM (excluding BCa and PCa), denosumab was non-inferior to ZA, but was not superior in delaying time to first and subsequent SREs [34]. However, an ad hoc analysis of this trial excluding the MM cohort was able to demonstrate a significant advantage of denosumab in delaying SREs [83]. There was no difference regarding OS or disease progression between patients treated with denosumab or ZA in each trial individually or in a combined analysis of the three trials (Table 2). 

An exploratory subgroup analysis of non-small cell lung cancer (NSCLC) patients from the phase 3 trial of denosumab versus ZA in the treatment of BM from solid tumors or MM suggested a significant OS advantage for denosumab [84]. However, in the recently published SPLENDOUR trial (NCT02129699), denosumab failed to show a measurable impact in OS, progression-free survival (PFS), or objective response rate (ORR) when added to standard first-line platinum-based doublet chemotherapy in advanced NSCLC, irrespective of the presence of BM at diagnosis or histological subtype [85].

In the case of MM, the Myeloma IX trial demonstrated that ZA has anti-myeloma effects beyond SRE prevention, evidencing a median PFS and OS improvement compared with clodronate [87]. Contrarily to the observed in NSCLC, the ad hoc subgroup analysis of MM patients from the phase 3 RCT comparing denosumab with ZA in the treatment of non-breast, non-prostate bone metastatic solid tumors and MM suggested a survival advantage for ZA over denosumab. However, since this trial was considered potentially confounded by imbalances in patient characteristics, antitumor therapies, and early withdrawals and limited by the small proportion of the MM cohort (10%) [88], a larger RCT focusing exclusively on MM patients was conducted [86]. This trial evidenced that denosumab was statically non-inferior in preventing SREs and carried a PFS but not an OS advantage compared with ZA. These results led to the approval of denosumab for the prevention of SREs in patients with MM, currently representing a particularly useful alternative in patients with renal dysfunction, a common clinical consequence in MM for which BPs may be contraindicated. 

In summary, a number of factors must be considered when selecting a BTA for bone health management in solid tumors or MM bone disease, namely drug availability, route of administration, and patient preference [80]. Denosumab seems to have an advantage over other BTAs due to its efficacy, convenience, and renal health benefits. However, BPs may be more cost-effective.

Discontinuation is another important aspect to consider regarding BTAs. While BPs incorporate into the bone matrix, having a prolonged action duration, denosumab has a short half-life and bone turnover suppression is not maintained after discontinuation. This justifies that the frequency of ZA administration may be reduced during disease remission periods or even that ZA is interrupted to allow safer dental treatments without substantially influencing the risk of SREs. Denosumab discontinuation, on the other hand, can result in rebound osteolysis that may lead to rapid bone loss, increased bone pain, and increased risk of SREs [89,90]. This supports the current recommendation to use BPs after stopping denosumab, as a way to minimize the clinical consequences of this rebound phenomenon [80]. 

#### 2.4.2. Prevention of BMs

As previously discussed, an improved understanding of the role of the RANKL/RANK pathway in cancer biology paved the way for the study of BTAs as a strategy to modify the course of primary cancers and possibly inhibit their metastatic spread, representing a promising opportunity for drug repurposing [91]. Many clinical trials accessing the disease-modifying properties of BTAs in human cancer have been conducted over the past two decades. RCTs comparing denosumab with ZA in bone health settings also studied disease-related outcomes, mostly as secondary endpoints, and represent a source of evidence of the disease-modifying properties of anti-RANKL therapy in advanced settings (Table 2). This knowledge was recently expanded in the SPLENDOUR trial, which studied the addition of denosumab to first-line platinum-based double chemotherapy in patients with stage IV NSCLC. Denosumab has also been studied in the adjuvant setting, with two placebo-controlled RCTs conducted on early BCa and one on high-risk non-metastatic CRPC (Table 3). The clinical evidence of denosumab in delaying or preventing bone metastatic disease is summarized next. 

##### Breast Cancer

BPs have been studied as a strategy to prevent the development of BM and disease recurrence in early-stage BCa for over 20 years, with initial studies reporting inconsistent results that were difficult to interpret. The clinical benefit of adjuvant BPs in early BCa only became clear when the 2015 Early BCa Trialists’ Collaborative Group (EBCTCG) large meta-analysis was published, showing that BPs reduced both BCa metastases in bone and deaths from BCa [94]. However, this benefit was limited to postmenopausal women or premenopausal women undergoing ovarian function suppression. Despite no formal approval in this indication, both ESMO and ASCO currently recommend the administration of BPs as adjuvant therapy in early BCa patients with low-estrogen status, particularly if deemed at high risk of relapse [95,96]. 

ABCSG-18 (NCT00556374) and D-CARE (NCT01077154) were two phase 3 RCTs that studied denosumab disease-modifying effects in the adjuvant setting of early BCa (Table 2). Their conflicting conclusions have recently been extensively discussed in the breast oncology community [68,92]. ABCSG-18 only included postmenopausal patients with hormone receptor (HR)+ early BCa under treatment with aromatase inhibitors (AIs) and used denosumab at a dose schedule approved for osteoporosis treatment. Previous preliminary results of this study had shown that adjuvant denosumab significantly delayed time-to-first clinical fracture, the trial’s primary endpoint [97]. Importantly, this reduction in fracture risk was irrespective of age or baseline bone mineral density. More recently, the authors reported an absolute DFS improvement of roughly 2% at 5 years and 3% at 8 years in the adjuvant denosumab group, which they recognized as modest but comparable to those from pivotal early BCa studies, considering the already favorable outcomes of this population with current standard treatment [68]. Importantly, most of the DFS benefit seen in the ABCSG-18 trial appeared to be related to a reduction in second non-breast primary cancers and deaths without recurrence, with little effect on histologically verified BCa recurrence, an observation that seems biologically difficult to interpret. D-CARE, on the other hand, included a broader and higher-risk BCa patient population (stages II or III; any type) receiving standard-of-care (neo)adjuvant systemic therapy and locoregional treatments. In this study, adjuvant denosumab at a more dose- and schedule-dense regimen than the one used in the ABCSG-18 trial failed to improve BM-free survival (BMFS), the primary study endpoint, as well as DFS or OS [92]. Furthermore, no subgroup appeared to experience a bone metastasis-free or DFS benefit, including HR+ postmenopausal patients. It is possible that the composite nature of the BMFS definition used in this trial might have diluted the effect of denosumab on bone recurrence, since approximately 40% of events contributing to this endpoint were deaths for any reason before patients developed BM. Likewise, the positive effect of denosumab on some exploratory bone-related endpoints, such as time to bone metastases as the site of first recurrence, might have been masked by other disease recurrence effects and thus did not lead to an overall improvement in clinical outcome. Notably, patients treated with denosumab in the D-CARE trial experienced a higher incidence of important adverse events, including osteonecrosis of the jaw.

Although it is unequivocal that the ABCSG-18 trial showed the benefit of denosumab on bone health of women under AIs, evidence for a disease-modifying effect in postmenopausal HR+ early BCa must be paralleled to that of the large EBCTCG meta-analysis showing the benefit of adjuvant BPs in BCa recurrence, distant recurrence, and BCa mortality in this same setting [94]. At this point, it is unclear whether current evidence on adjuvant denosumab is enough to change international recommendations on the use of adjuvant BPs in early BCa. Although ABCSG-18 authors claim that women should ultimately be empowered to choose between adjuvant BP and adjuvant denosumab therapy, it is not completely clear that both options are truly interchangeable when it comes to efficacy. The apparent differences between D-CARE and EBCTCG meta-analysis results suggest that the benefits of adjuvant BPs may not simply reflect their primary effect on bone cell function, but arise from other additional modulatory effects on BCa metastatic process, as previously discussed. Considering the lack of comparative evidence, further studies are required, ideally comparing both strategies in the same study population. Ongoing translational studies and exploratory analyses based on tumor samples from these trials may allow the identification of clinically useful predictive biomarkers in the future, allowing to select patients more likely to respond to denosumab therapy. Until more data become available, given the greatest body of evidence and according to international treatment guidelines, BPs should remain the BTA of choice as a disease-modifying agent in early BCa postmenopausal women.

##### Prostate Cancer

Although the role of BTAs in the prevention of BM from PCa has been extensively addressed, all clinical trials investigating the efficacy of adjuvant BPs in men with PCa failed to demonstrate a disease recurrence or metastases prevention benefit [98,99]. On the other hand, in a phase 3 RCT conducted in men with high-risk non-metastatic CRPC, denosumab increased median BMFS by 4.2 months over placebo and significantly delayed time to first BM, despite no OS advantage [93]. This trial was a proof-of-concept and highlighted the clinical relevance of bone microenvironment and RANKL/RANK signaling in PCa relapse in bone. However, these disease benefits were not deemed sufficient to justify the 5% cumulative incidence of osteonecrosis of the jaw observed in the denosumab treatment arm and this intervention was not granted regulatory approval.

## 3. RANKL/RANK Pathway in Breast: Friend and Foe

As discussed in the previous section, the RANKL/RANK pathway is also an intrinsic characteristic of several tumors, assuming particular relevance in BCa, where it was found to have a major role in breast physiology and carcinogenesis (Figure 3). This section will review the pre-clinical evidence that led to the discovery of RANKL-mediated breast carcinogenesis and characterization of RANK-expressing BCa, as well as the clinical evidence of RANKL inhibition in BCa bone metastatic disease treatment and prevention.

### 3.1. Breast Carcinogenesis

The RANKL/RANK pathway was implicated in breast physiology following observations that mice lacking RANKL or RANK were unable to form lobulo-alveolar mammary structures during pregnancy, due to lack of RANKL autonomous effect on epithelial cells [100]. Subsequently, it was shown that RANKL controls mammary epithelial cell proliferation via IKKα-cyclin D1 [101] and that the expansion of mammary stem cells was mediated by paracrine RANKL signaling, which is expressed by luminal cells under progesterone receptor (PR) control [7,102,103]. Accordingly, administration of medroxyprogesterone acetate (MPA) progestin was able to induce RANKL expression in mammary epithelial cells, whereas RANK genetic inactivation was able to abrogate MPA-induced epithelial proliferation and stem cell expansion, with a significant decrease in the incidence of MPA-driven mammary cancer, which onset was delayed [7]. In another study, RANKL ablation in mammary epithelium blocked progesterone-induced morphogenesis, and systemic RANKL administration was able to trigger the proliferation of mammary cells in absence of PR signaling [102]. Importantly, progesterone-induced carcinogenesis was abrogated using a RANKL inhibitor [7,103]. Overall, this evidence suggested a link between RANKL/RANK pathway and breast carcinogenesis, including a short-term increase in BCa incidence during pregnancy, and suggested that RANKL could be targeted to prevent BCa. In accordance with these findings, a study comparing pregnant and matched young BCa patients showed that the expression of RANKL, but not RANK, was more prevalent in the pregnant group, both in tumor and adjacent normal tissue [104]. RANKL expression was significantly higher in PR-positive and luminal A-like tumors, whereas RANK expression was higher in triple-negative tumors. 

The hypothesis of using RANKL inhibition to prevent BCa was extended to BRCA-mutated BCas, following observations that the effects of RANK genetic inactivation in the mammary epithelium on BCa onset and incidence were also observed in a Brca1-p53 mutated mouse model, and that RANKL pharmacological inhibition abolished pre-neoplastic lesions [105]. It was shown that RANKL/RANK controlled proliferation and expansion of Brca1-p53 mutant mammary stem cells and that genome variations within *RANK* locus were significantly associated with the risk of developing BCa in women with BRCA1 mutations. Another study investigated the role of the RANKL/RANK pathway in the pre-neoplastic phase of BRCA1 mutation carriers and showed that RANK+ luminal progenitors were highly proliferative, with aberrant DNA repair and a molecular signature of basal-like BCa [106]. In this study, RANKL inhibition was effective in three-dimensional breast organoids derived from pre-neoplastic BRCA1mut/+ tissue and in breast biopsies from BRCA1 mutation carriers. Using isogenic pairs of normal-like human breast epithelial cells in which inactivation of a single BRCA1 allele results in genomic instability, it was also shown that BRCA1 haploinsufficiency upregulated RANKL in vitro, and that neutralizing RANKL abrogated the formation of mammospheres [107]. Additionally, it was also reported that circulating OPG levels were lower among women with inherited BRCA1 mutations compared to women with baseline population risk, particularly in BRCA1-mutated ones [108], as well as a significant inverse relation between circulating OPG levels and BCa risk among women with BRCA1/2 mutations [103]. 

### 3.2. Prognosis

As previously referred, different cancer cell types express RANK and are responsive to RANKL in vitro [109]. A first retrospective analysis of RANK expression assessed by immunohistochemistry (IHC) in 74 BM tissues from solid tumors, including breast, colorectal, renal, lung, and PCa, showed that 89% of BM were RANK+ and the median percentage of RANK+ cells was not different in paired primary tumors [110]. This was the first evidence in human tissue that RANKL inhibition could also be used in the adjuvant setting to directly target cancer cells. In human BCa, several subsequent studies correlated the expression of RANKL, RANK, and/or OPG with prognosis. 

In a cohort of 295 primary BCa patients, lower *RANK* and high *OPG* mRNA levels correlated with longer OS and DFS, and RANK detection by IHC was associated with BM development and shorter skeletal DFS, with RANK-negative and RANK+ patients displaying shorter skeletal DFS of 105.7 months and 58.9 months, respectively [111]. In another cohort study of 102 patients with metastatic BCa, RANK expression assessed by IHC disclosed 47.1% of RANK+ cases with significantly poor PFS and disease-specific survival, but only in the BM group [112]. 

Contradictory results were reported in a study where *RANKL*, *RANK*, and *OPG* were assessed by RT-qPCR in a cohort of 127 primary BCa tissue samples and 31 matching non-neoplastic mammary samples [113]. In this study, *RANK*, *RANKL,* and *OPG* transcript levels were shown to be reduced in tumor samples versus normal tissue. Lower RANK and RANKL expression were associated with worse clinical outcomes, and lower OPG expression levels were associated with significantly better OS. 

A different study investigated the correlation between RANKL and RANK IHC expression and pathological complete response (pCR), DFS, and OS in BCa patients from the neoadjuvant GeparTrio study [114]. RANK expression was reported in 160 (27%) patients and correlated with high tumor grade, negative HR status, higher pathological complete response rate, and shorter DFS and OS, suggesting an association with higher chemotherapy sensitivity but also with higher relapse and death risk. RANKL expression was only observed in 6% of cases. A contemporary study reported a higher percentage of RANK and RANKL expression in BCa tissues, with 74.1% and 78.4% of RANK+ and RANKL+ cases, respectively, amongst 185 samples [115]. No significant differences were found regarding clinicopathologic features between tumors with or without RANK, although RANK expression was significantly associated with poor DFS. 

Currently, RANKL and RANK staining by IHC remains controversial, and differences observed in these studies most likely reflect methodological aspects, like the use of unspecific antibodies. Evidence regarding OPG is contradictory and associates OPG expression with both good and poor survival outcomes [116]. The notion that OPG expression in tumors is inversely correlated with tumor grade seems to be unanimous.

### 3.3. Aggressiveness 

RANK pathway mediates the acquisition of cellular features related to aggressiveness and invasiveness. In RANK+ cells, RANKL triggers activation of a downstream transduction cascade involving multiple pathways, like NF-kB, AKT/PKB, JNK, ERK, Src, and MAP kinase cascade [38,39]. RANK-induced cellular features include increased migration and invasion, stemness and transformation, epithelial–to–mesenchymal transition (EMT), anchorage-independent growth, and metastatic ability [37,38,39,43,117,118,119,120]. 

Together with clinical evidence, most studies linked the deleterious effect of RANK expression to triple-negative BCa (TNBCa) cell lines. However, as previously mentioned, also human ER+ BCa cases were found to be RANK+ [114]. Nevertheless, the biological implications of RANK expression in ER+ BCa have remained elusive until recently. Our group has shown that ER+/HER2- RANK-overexpressing BCa cells have a staminal and mesenchymal phenotype, with decreased proliferation rate and decreased chemotherapy and ET susceptibility [47]. Interestingly, RANK-overexpressing cells showed a remarkably low proliferation rate in vivo, which may be associated with intrinsic therapy resistance. This phenotype could be linked to human luminal A BCa, since in silico analysis of the transcriptome of human breast tumors confirmed the association between RANK expression and stem cell and mesenchymal markers and predicted decreased proliferation index in this setting. Moreover, continuous RANK pathway activation with RANKL was able to downregulate HR in vitro and in vivo and increased ET resistance independently of RANK levels. Recently, it was demonstrated that in non-transformed mammary epithelia, RANK ectopic expression was associated with oncogene-induced senescence, including reduced proliferation, increased senescence-associated β-galactosidase activity, and dependency on p16/p19 tumor suppressors. These features initially delayed tumor onset in oncogene-driven models, like Neu and PyMT, but promoted stemness tumor growth and metastases [121].

Additionally, RANK and HER2 pathways seem to modulate HER2-driven carcinogenesis [122]. HER2 activates NF-κB via IKKα, promoting tumor progression in ER-negative/HER2+ BCa cells in response to RANK signaling [7,123]. It was recently demonstrated that RANK and RANKL were more frequent in HER2+ tumors with acquired resistance to anti-HER2 therapies, and that RANK expression increased after dual anti-HER2 neoadjuvant therapy in the SOLTI-1114 PAMELA trial cohort [124]. In this study, it was shown that RANKL stimulation increased NF-κB activation in lapatinib-resistant HER2+ cell lines compared to their sensitive counterparts, whereas RANK loss sensitized lapatinib-resistant cells to the drug in vitro. Furthermore, it was also shown that RANK binds to HER2 in BCa cells and that enhanced RANK pathway activation alters HER2 phosphorylation status, supporting a link between RANK and HER2 signaling in BCa.

Overall, extensive pre-clinical evidence clearly suggests a role for RANKL inhibition in different BCa stages, from prevention to metastatic treatment. In the next section, clinical findings of RANKL inhibition in the prevention of BRCA-driven BCa and neoadjuvant therapy will be addressed.

### 3.4. Therapeutic Perspectives of RANKL Inhibition in Early BC beyond BM Prevention

After evidence of the potential therapeutic relevance of RANKL/RANK pathway blockade beyond cancer therapy-induced bone loss and BM management, several ongoing and completed clinical trials aimed to evaluate the anti-tumor effect of denosumab in BCa prevention and neoadjuvant treatment.

#### 3.4.1. Prevention (BRCA-Mutated BCa)

Denosumab is being investigated in the prevention of BCa in BRCA1 mutation carriers. In this context, denosumab could delay the need for bilateral prophylactic mastectomy and counteract bone loss following bilateral prophylactic salpingo-oophorectomy. BRCA-D (ACTRN12614000694617), a pre-operative, proof-of-concept pilot study, is exploring whether short-term treatment with denosumab affects Ki67 expression and other interesting biological markers in the normal breast tissue of women carrying BRCA1 or BRCA2 mutations. Denosumab as chemoprevention in germline BRCA1 mutation carriers will also be the subject of a phase 3, placebo-controlled RCT planned to start in 2021, entitled BRCA-P (NCT04711109). 

Finally, considering that RANK expression has been reported in a significant proportion of cancers arising in BRCA1 mutation carriers [105], it will also be interesting to study denosumab in the early stages of BRCA1-driven tumorigenesis and/or as add-on therapy to reduce the risk of contralateral BCa.

#### 3.4.2. Neoadjuvant Treatment

Several prospective, mostly phase 1 or 2, trials are also investigating the impact of neoadjuvant denosumab treatment on BCa local biological characteristics, such as tumor proliferation, apoptosis, or immune tumor microenvironment. These studies usually compare tumor features from a baseline biopsy with post-denosumab administration surgical specimens. One of these trials, D-BEYOND (NCT01864798), accessed the effects of a short, pre-surgical course of denosumab in premenopausal women with early BCa [10]. The study’s primary endpoint, the geometric mean decrease in the percentage of Ki67+ cells, was not significant, neither was tumor cell survival accessed by cleaved caspase-3. However, this trial disclosed promising immunomodulatory properties for denosumab, which will be discussed in more detail in the next section. A similar and recently completed trial (NCT02900469) aimed to determine the impact of presurgical denosumab on pharmacodynamic markers of RANKL inhibition. D-BIOMARK (NCT03691311), an ongoing early phase 1 study, will measure neoadjuvant denosumab antiproliferative and pro-apoptotic activity and correlate it with RANKL and RANK tumor expression, BCa phenotype, and patient’s menopausal status. 

Differently from the previously mentioned studies, GeparX (NCT02682693), a recently terminated phase 2 trial that enrolled 780 BCa patients, evaluated whether the addition of denosumab to anthracyclin/taxan-containing neoadjuvant chemotherapy increased pCR rate and improved outcomes regarding RANK tumor expression. Study results are currently awaited. 

## 4. RANKL/RANK Pathway as a Mediator of Systemic and Tumor Microenvironment (Innate and Acquired) Immunity 

As discussed above, RANKL/RANK pathway is an important mediator of immune cell activity. However, in patients treated with denosumab in the context of BM, no clinically significant effects were observed in the immune system, probably due to the non-essential nature of RANKL. Yet, in the past decade, preclinical evidence suggested that RANKL/RANK pathway inhibition could be a potential strategy to improve the effectiveness of immune checkpoint inhibitors (ICI) in cancer treatment [125].

### 4.1. Breast Cancer

Following the pre-clinical observation that IKKα activation by RANKL was correlated with metastatic progression of PCa and tumor infiltration with RANKL-expressing inflammatory cells [126], the HER2-induced mammary carcinogenesis model was subsequently used to address RANKL nature [127]. In this model, RANK signaling was associated with pulmonary metastases and CD4+CD25+FoxP3+T cell dependency, the major pro-metastatic function of which appeared to be RANKL production [127]. Interestingly, T cell-dependence of pulmonary metastases was replaced by the administration of exogenous RANKL, which also stimulated pulmonary metastases of RANK+ human BCa cells. As tumor-infiltrating CD4+ or FoxP3+ T cells are associated with poor prognosis in human BCa, this suggested that RANKL inhibition could be used in combination with other therapies to improve outcomes, including with ICIs, which are associated with substantially disparate responses in different tumor types and patients [128].

In human BCa, mostly refractory to ICIs, RANKL was observed in tumor-infiltrating lymphocytes (TILs) and RANK was strongly expressed in TAMs [7]. Therefore, RANKL is a chemoattractant for TAMs, and RANKL/RANK signaling in M2 macrophages promotes proliferation of Treg lymphocytes, leading to an immunosuppressive environment [127]. In BCa clinical setting, although denosumab failed to reduce tumor proliferation and tumor cell survival in the single-arm, neoadjuvant, phase 2 D-BEYOND trial (NCT01864798), it significantly increased stromal and intratumoral lymphocyte levels, with 45.8% of tumor samples showing a ≥10% increase in stromal TILs, the study’s cutoff for patients to be considered responders [10]. Using a murine model of ER+ BCa, loss of RANK signaling in tumor cells was shown to increase leukocytes, lymphocytes, and CD8+ T cells and to reduce immunosuppressive macrophage and neutrophil infiltration [10], again suggesting that RANKL inhibition may increase the anti-tumor effect of immunotherapies in BCa through a tumor cell-mediated effect. This was confirmed in the D-BEYOND cohort, where higher RANK signaling activation in tumors, higher soluble RANKL serum levels, and a higher percentage of intratumoral Tregs were predictive of denosumab-induced immunomodulatory response. The biomarker study D-BIOMARK will help to clarify if the immune response seen in the D-BEYOND trial is also observed in postmenopausal women. 

### 4.2. Melanoma and Non-Small Cell Lung Cancer

In parallel with promising pre-clinical evidence, a case report described an unexpected dramatic partial response in a patient with rapidly advancing bone metastatic melanoma after concomitant treatment with denosumab and the anti-CTLA-4 antibody ipilimumab, suggesting that the combination of both agents could have a synergistic effect [129]. This synergistic effect in metastatic melanoma was subsequently explored in a retrospective analysis. Although the authors did not observe a statistically significant OS, PFS, or ORR advantage in the cohort treated with the combination, patients receiving ICIs and denosumab did not behave poorly, despite having poorer prognostic features [8]. Furthermore, real-world evidence from an observational study in patients with advanced melanoma or NSCLC found that a longer mean duration of concomitant ICI and denosumab therapy was associated with improved overall response in both tumor types and increased OS in NSCLC [130]. Nevertheless, due to the complexity of interactions between tumor cells and the immune system, the relevance of RANK signaling will depend on both tumor and microenvironment.

### 4.3. (Immuno)Therapeutic Perspectives of RANKL Inhibition 

Several ongoing prospective clinical trials are exploring the combination of denosumab with ICIs in different human cancers (Table 4). 

CHARLI (NCT03161756) is a phase 1b/2 trial investigating denosumab in combination with the anti-PD1 nivolumab, with or without ipilimumab, in metastatic melanoma. The POPCORN trial (ACTRN12618001121257) will evaluate immune changes in NSCLC patients treated with nivolumab alone or in combination with denosumab. KEYPAD (NCT03280667), a phase 2 trial, will study the combination of denosumab with the PD-1 inhibitor pembrolizumab in VEGF receptor inhibitor-refractory clear-cell RCC. Future prospective trials ascertaining the disease-modifying relevance and immunomodulatory properties of denosumab in patients with BCa should also be encouraged. 

## 5. Conclusions and Future Perspectives

Over the last few years, our knowledge about the RANKL/RANK pathway has expanded, and RANKL targeting potential is now under study across the different stages of cancer progression, in different types of cancer, and with different purposes, from the prevention of BRCA-mutated BCa to the combination with targeted therapies, including ICIs, in metastatic BCa, melanoma and NSCLC. 

Compelling evidence suggests that targeting RANKL may potentially contribute to change the paradigm of BCa treatment in different disease stages. Facts have also started to accumulate showing that the relevance of the RANKL/RANK pathway is not restricted to TNBCa. This opens the possibility of combining RANKL/RANK pathway inhibitors with ET and anti-HER2 or other targeted therapies to improve efficacy and overcome resistance.

The potential of RANKL inhibition to leverage the efficacy of ICIs in melanoma and NSCLC is exciting and introduces another perspective to the pan-cancer research on RANKL/RANK pathway. However, unlike these highly immunogenic solid tumors, BCa responsiveness to immunotherapy is scarce and this strategy is currently only recommended in advanced PD-L1-positive TNBCa [131,132]. Although TILs are prognostic and predictive in HER2+ and TNBCa [133], they are infrequent in most luminal breast tumors, emphasizing the relevance of a tumor microenvironment-changing therapy able to sensitize these tumors to ICIs. Recently published preclinical findings and data derived from D-BEYOND clinical trial suggest that denosumab may have immunomodulatory properties that can assume a particularly relevant role in poorly-infiltrated, immunotherapy-insensitive luminal BCas [10]. These findings should encourage future prospective trials accessing the clinical relevance of combining denosumab with ICIs in BCa, similarly to those currently ongoing in melanoma and NSCLC.

Overall, the resurging of pre-clinical and clinical research on the RANKL/RANK pathway has the potential to translate into effective treatments, with an impact across different cancer types. Undoubtedly, several topics remain to be explored in forthcoming years. 

## Figures and Tables

**Figure 1 cells-10-01978-f001:**
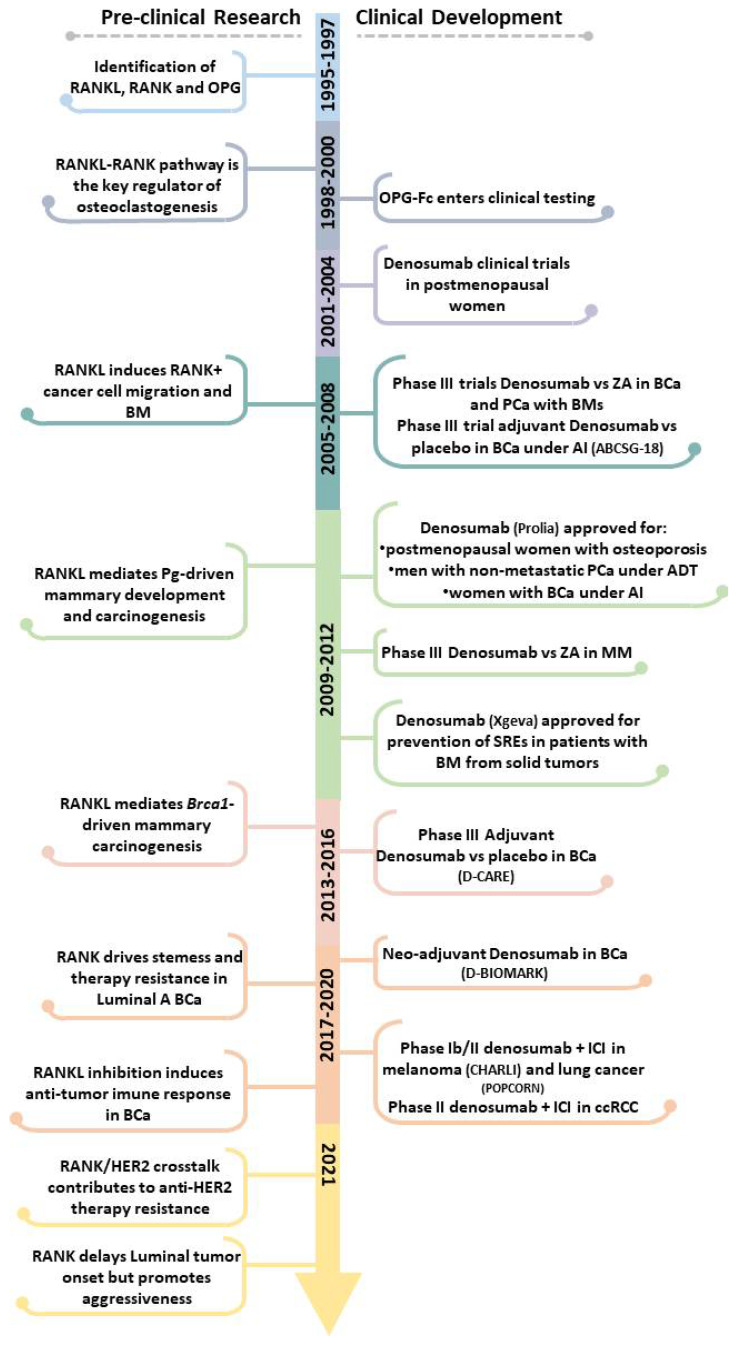
Pre-clinical and clinical landmarks of RANKL/RANK pathway research in Oncology. ADT, androgen deprivation therapy; AI, aromatase inhibitors; BCa, breast cancer; BM, bone metastases; ccRCC, clear cell renal cell carcinoma; ICI, immune checkpoint inhibitor; MM, multiple myeloma; PCa, prostate cancer; Pg, progesterone; SREs, skeletal-related effects; ZA, Zoledronate.

**Figure 2 cells-10-01978-f002:**
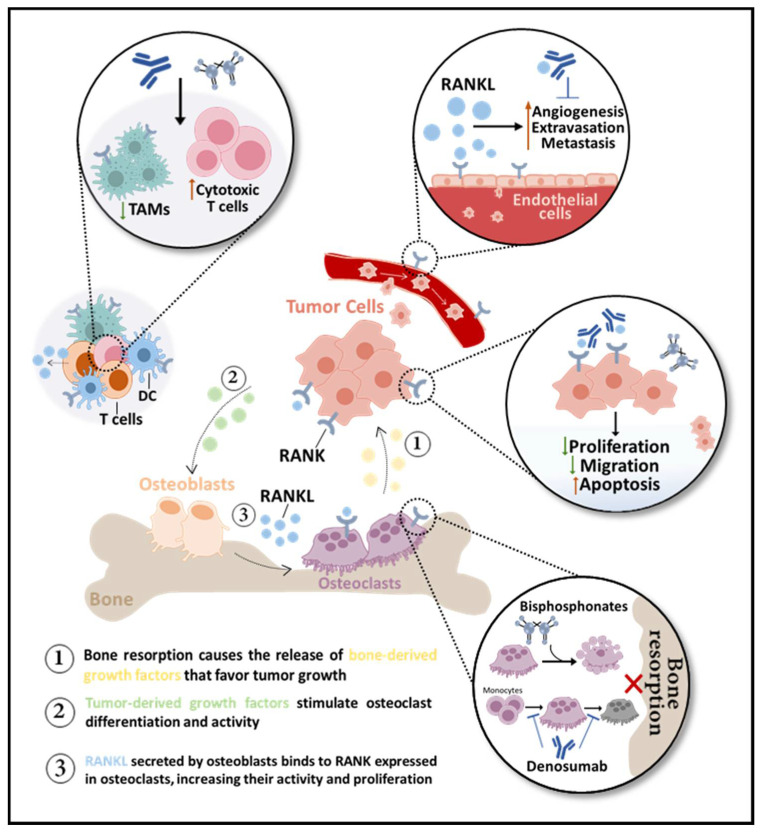
RANKL inhibition in bone metastases (BM). Bone-targeted agents (BTAs) are used to control BM by impairing bone resorption, indirectly affecting the tumor burden. While bisphosphonates only affect mature osteoclasts, inducing apoptosis, the anti-RANKL antibody denosumab prevents osteoclast differentiation, activity, and survival. Denosumab may also be antiangiogenic over RANK-positive endothelial cells. RANK is expressed in dendritic cells (DC) and macrophages, like tumor-associated macrophages (TAMs), and T-cell derived RANKL inhibition decreases TAMs and increases cytotoxic T cells. Potential direct effects of BTAs in tumor cells include inhibition of proliferation and migration and induction of apoptosis.

**Figure 3 cells-10-01978-f003:**
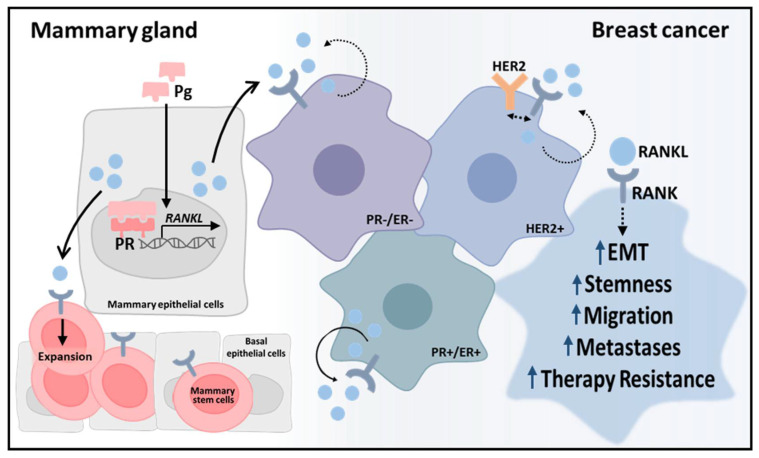
The role of RANKL/RANK signaling in the breast spans from physiologic development to breast cancer (BCa). RANKL expression in mammary epithelial cells is regulated by progesterone (Pg)-activated receptors and activates RANK in RANK-expressing luminal and basal cells, promoting proliferation, differentiation, migration, and survival. Expansion of mammary stem cells is associated with carcinogenesis. In BCa cells, RANKL-RANK signaling pathway is associated with aggressiveness features independently of subtype, including epithelial-to-mesenchymal transition (EMT), stemness, migration, metastases and therapy resistance.

**Table 1 cells-10-01978-t001:** Head-to-head randomized controlled trials comparing denosumab with zoledronate for delay or prevention of skeletal-related events in bone metastatic solid tumors and multiple myeloma.

Cancer Type(s)	First On-Study SREs (% of Patients; D vs. ZA)	Time to First SRE	Time to First and Subsequent SREs	Ref.
Breast (*n* = 2046)	NE	Denosumab superior (HR 0.82; 95% CI 0.71–0.95; *p* < 0.001 NI; *p* = 0.01 S)	Denosumab superior (RR 0.77; 95% CI 0.66–0.89; *p* = 0.001 S)	[32]
CRPC (*n* = 1901)	36 vs. 41	Denosumab superior (HR 0.82; 95% CI 0.71–0.95; *p* = 0.0002 NI; *p* = 0.008 S)	Denosumab superior (RR 0.82; 95% CI 0.71–0.94; *p* = 0.008)	[33]
Solid tumors (excluding breast and prostate) and MM (*n* = 1779)	NE	Denosumab non-inferior, but not statistically superior (HR 0.84; 95% CI, 0.71 to 0.98; *p* = 0.0007 NI; *p* = 0.06 S)	Denosumab not statically superior (RR 0.90; 95% CI 0.77–1.04; *p* = 0.14)	[86]
MM (*n* = 1718)	44 vs. 45	Denosumab non-inferior, but not statistically superior (HR 0.98; 95% CI 0.85–1.14; *p* = 0.01 NI)	Denosumab not statically superior (RR 1.01; 95% CI 0.89–1.15; *p* = 0.84)	[86]

CI, Confidence Interval; CRPC, Castration-Resistant PCa; D, Denosumab; HR, Hazard Ratio; MM, Multiple Myeloma; NE, Not Evaluable; NI, Non-Inferiority; RR, Rate Ratio; S, Superiority; SREs, Skeletal-Related Events; ZA, Zoledronate.

**Table 2 cells-10-01978-t002:** Randomized controlled trials of denosumab disease-modifying properties in advanced human cancer.

Cancer Type(s)	Number of Patients	Intervention	Disease-Related Outcomes	Trial Identifier/Reference
Breast (advanced, all types, pre-and postmenopausal)	2046	Denosumab vs. ZA	Similar OS (HR 0.95; 95% CI 0.81–1.11; *p* = 0.49) and time to disease progression (HR 1.00; 95% CI 0.89–1.11; *p* = 0.93).	NCT00321464 [32]
CRPC	1901	Denosumab vs. ZA	Similar OS (HR 1.03; 95% CI 0.91–1.17; *p* = 0.65) and time to disease progression (HR 1.06; 95% CI 0.95–1.18; *p* = 0.30).	NCT00321620 [33]
Solid tumors (excluding breast and prostate) and MM	1779	Denosumab vs. ZA	Similar OS (HR 0.95; 95% CI 0.83–1.08; *p* = 0.43) and time to disease progression (HR 1.00; 95% CI 0.89–1.12; *p* = 1.00). Ad hoc analyses favored denosumab for NSCLC patients (HR 0.79; 95% CI 0.65–0.95) and ZA for MM patients (HR 2.26; 95% CI 1.13–4.50).	NCT00330759 [34]
NSCLC (stage IV)	514	ChT + Denosumab vs. ChT	Similar OS (HR 0.96; 95% CI 0.78–1.19; *p* = 0.36), PFS (HR 0.99; 95% CI 0.82–1.19; *p* = 0.46) and ORR (30.5% vs. 29.4%; *p* = 0.85).	NCT02129699 (SPLENDOUR) [85]
MM	1718	Denosumab vs. ZA	Denosumab improved PFS by 10.7 months (HR, 0.82; 95% CI 0.68–0.99; *p* = 0.036). Similar OS (HR, 0.90; 95% CI 0.70–1.16; *p* = 0.41).	NCT01345019 [86]

ChT, Chemotherapy; DFS, Disease-Free Survival; MM, Multiple Myeloma; NCT, National Clinical Trial; NSCLC, Non-Small Cell Lung Cancer; ORR, Objective Response Rate; OS, Overall Survival; PFS, Progression-Free Survival; ZA, Zoledronate.

**Table 3 cells-10-01978-t003:** Randomized clinical trials of adjuvant denosumab disease-modifying properties in early human cancer.

Cancer Type(s)	Number of Patients	Intervention	Disease-Related Outcomes	Trial Identifier/Reference
Breast (adjuvant, early-stage, ER+, posmenopausal, under AIs)	3425	Denosumab vs. Placebo	Denosumab increased 5-year DFS by 1.9% and 8-year DFS by 3.1% (HR 0.82; 95% CI 0.69–0.98; *p* = 0.0260).	NCT00556374 (ABCSG-18) [68]
Breast (adjuvant, stage II-III, all types, high-risk, pre-and postmenopausal)	4509	Denosumab vs. Placebo	Similar BMFS (HR 0.97; 95% CI 0.82–1.14; *p* = 0.70), DFS (HR 1.04; 95% CI 0.91–1.19; *p* = 0.57), DRFS (HR 1.06; 95% CI 0.92–1.21; *p* = 0.41) and OS (HR 1.03; 95% CI 0.85–1.25; *p* = 0.76).	NCT01077154 (D-CARE) [92]
CRPC (high-risk, non-metastatic)	1432	Denosumab vs. Placebo	Denosumab improved BMFS by 4.2 months (HR 0.85; 95% CI 0.73–0.98; *p* = 0.028) and delayed time to first BM (HR 0.84; 95% CI 0.71–0.98; *p* = 0.032). Similar OS (HR 1.01; 95% CI 0.85–1.20; *p* = 0.91).	NCT00286091 [93]

AIs, Aromatase Inhibitors; BM, Bone Metastases; BMFS, Bone Metastases-Free Survival; DFS, Disease-Free Survival; DRFS, Distant-Recurrence-Free Survival; ER, Estrogen Receptor; NCT, National Clinical Trial; OS, Overall Survival.

**Table 4 cells-10-01978-t004:** Clinical trials investigating the immunomodulatory properties of denosumab in combination with immune checkpoint inhibitors in human cancer.

Phase	Cancer Type	Intervention	Primary Endpoint	Other Endpoints	Status	Trial Identifier
**1b/2**	Melanoma (unresectable, stage III/IV)	Ipilimumab+ Nivolumab+Denosumab vs. Nivolumab + Denosumab	PFS, grade 3–4 irAEs	OS	Recruiting	NCT03161756 (CHARLI)
**1b/2**	NSCLC (neoadjuvant, resectable, stage Ia-IIIa)	Nivolumab + Denosumab vs. Nivolumab	TCR clonality, RNA/transcription profile and genomic changes, markers of interest (IHC)	MPR, rate of R0 resection, radiological response, PFS, OS	Recruiting	ACTRN12618001121257 (POPCORN)
**2**	Renal (ccRCC, advanced, refractory to VEGFR-TKIs)	Denosumab + Pembrolizumab (single-arm)	OTR	PFS, time to OTR, DCR, time to first SRE	Recruiting	NCT03280667 (KEYPAD)

ACTRN, Australian New Zealand Clinical Trials Registry; ccRCC, Clear-Cell Renal Cell Carcinoma; DCR, Disease-Control Rate; IHC, immunohistochemistry; irAEs, immune-related Adverse Events; MPR, Major Pathological Response; NCT, National Clinical Trial; NSCLC, Non-Small Cell Lung Cancer; OS, Overall Survival; OTR, Objective Tumor Response; PFS, Progression-Free Survival; SRE, Skeletal-Related Event; TCR, T-Cell Receptor; VEGFR-TKIs, Vascular Endothelial Growth Factor Receptor Tyrosine Kinase Inhibitors.

## Data Availability

Not applicable.

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
