# Peer review of "The Roadmap of RANKL/RANK Pathway in Cancer"

_cells, 2021, doi:10.3390/cells10081978_

Round 1

Reviewer 1 Report

Manuscript Cells-1308357 

Reviewer comments

The manuscript entitled " The roadmap of RANKL/RANK pathway in cancer " by Sandra Casimiro et al. is a review treating parts of RANKL/RANK pathway in bone health and disease more specifically cancers with metastases to bone. The manuscript is well organized and written, presenting the various implications of this pathway in the different processes subjacent to tumor growth and dissemination to bone. Clinical trials based on the evaluation of inhibitors of this pathway are also presented what make of this manuscript a completed review.

The Reviewer has no comment on this very interesting manuscript but suggests an edition as few errors of English usage are present.

Author Response

Dear Reviewer we appreciate your very positive comments and took this opportunity to fully review our manuscript in terms of style, grammar and spelling. 

Reviewer 2 Report

This is an exaustive review of the literature concerning a relevant pathway in tumor progression and above all an effective target to block bone metastasis. The deep revision of the clinical trials performed or ongoing is really appreciable and useful also for clinicians and not only for researchers. The manuscript is suitable for publication.

Author Response

(The authors gave the same response as above.)

Reviewer 3 Report

This is a well-written review paper. The following concerns should be clarified before publication.

  1. There are too many abbreviations in this manuscript. Please add a list of abbreviations for those who are not so familiar with the abbreviations for frequent check.
  2. There are many tumor origins in this manuscript. The authors should add subtitle of each tumor origin. For example, 2.4.1. Bone metastatic disease can be considered add subtitle of each tumors. Same as 2.4.2. Prevention of BMs.
  3. “3. RANKL/RANK pathway in breast: Friend and foe.” The authors should consider to rephrase as: 3. RANKL/RANK pathway: Friend and foe. 3.1 Breast, 3.2 xx cancers or other cancers.
  4. “4. RANKL/RANK pathway as mediator of systemic and tumor microenvironment (innate and acquired) immunity” The authors should add subtitle of each tumor origin.
  5. “5. Conclusions and future perspectives” which mainly focus on breast cancer. The authors should include all kind of cancers mentioned in the manuscript.
  6. If possible, the author can add an carton figure of the role of RANKL/RANK signaling in caners other than breast cancer like figure 3.

Author Response

ear Reviewer we appreciate your comments and took this opportunity to fully review our manuscript in terms of style, grammar and spelling. We considered all your suggestions for the improvement of the manuscript, and sincerely hope you find the revised version suitable for publication.

  1. There are too many abbreviations in this manuscript. Please add a list of abbreviations for those who are not so familiar with the abbreviations for frequent check.

A comprehensive list of abbreviations is now included. We also removed several abbreviations that were unnecessary.

  1. There are many tumor origins in this manuscript. The authors should add subtitle of each tumor origin. For example, 2.4.1. Bone metastatic disease can be considered add subtitle of each tumors. Same as 2.4.2. Prevention of BMs.

We agree that by focusing in several cancer types there are several sections that present data from different tumors without a formal separation. However, the vast majority of clinical trials of RANKL inhibition in bone metastases treatment included patients with bone metastases from different types of solid tumors and multiple myeloma. This would imply to repeat the trial in different sub-sections when our purpose is actually to show the transversal nature of the findings. In the topic of prevention of bone metastases (2.4.2.), although the majority of data comes from breast cancer studies, we could sub-title breast and prostate cancer without breaking the rational.

  1. “3. RANKL/RANK pathway in breast: Friend and foe.” The authors should consider to rephrase as: 3. RANKL/RANK pathway: Friend and foe. 3.1 Breast, 3.2 xx cancers or other cancers.

This section of the manuscript is fully dedicated to breast cancer, although we present pre-clinical studies that used models of other cancer types in 3.2. and 3.3. We feel that a formal separation of these few examples would break the text and rational, so we opted to maintain the original arrangement.

  1. “4. RANKL/RANK pathway as mediator of systemic and tumor microenvironment (innate and acquired) immunity” The authors should add subtitle of each tumor origin.

In section 4 we introduced a formal separation between breast cancer and melanoma/NSCLC.

  1. “5. Conclusions and future perspectives” which mainly focus on breast cancer. The authors should include all kind of cancers mentioned in the manuscript.

We agree that the Conclusions were mostly focused on breast cancer, which derives from the vast majority of studies focusing on this disease. We revised the Conclusions to make clear the relevance of RANKL/RANK pathway in the different types of cancer mentioned throughout the manuscript.

  1. If possible, the author can add a carton figure of the role of RANKL/RANK signaling in cancers other than breast cancer like figure 3.

We appreciate your suggestion. We feel that at this stage there is not enough background to illustrate the effects of RANK signaling in other cancers like in breast cancer (Figure 3). Nevertheless, we extensively reviewed the literature and are confident that all relevant findings in all cancers were included through the text.